# Mapping nucleolus-associated chromatin interactions using nucleolus Hi-C reveals pattern of heterochromatin interactions

Ting Peng[1,9], Yingping Hou[1,9], Haowei Meng [1,9], Yong Cao [2], Xiaotian Wang[1,3], Lumeng Jia[1], Qing Chen [4], Yang Zheng[5], Yujie Sun [3,6], Hebing Chen [5], Tingting Li [7] ✉ & Cheng Li [1,8] ✉

As the largest substructures in the nucleus, nucleoli are the sites of ribosome biogenesis. Increasing evidence indicates that nucleoli play a key role in the organization of 3D genome architecture, but systematic studies of nucleolus-associated chromatin interactions are lacking. Here, we developed a nucleolus Hi-C (nHi-C) experimental technique to enrich nucleolus-associated chromatin interactions. Using the nHi-C experiment, we identify 264 high-confidence nucleolus-associated domains (hNADs) that form strong heterochromatin interactions associated with the nucleolus and consist of 24% of the whole genome in HeLa cells. Based on the global hNAD inter-chromosomal interactions, we find five nucleolar organizer region (NOR)-bearing chromosomes formed into two clusters that show different interaction patterns, which is concordant with their epigenetic states and gene expression levels. hNADs can be divided into three groups that display distinct *cis/trans* interaction signals, interaction frequencies associated with nucleoli, distance from the centromeres, and overlap percentage with lamina-associated domains (LADs). Nucleolus disassembly caused by Actinomycin D (ActD) significantly decreases the strength of hNADs and affects compartment/TAD strength genome-wide. In summary, our results provide a global view of heterochromatin interactions organized around nucleoli and demonstrate that nucleoli act as an inactive inter-chromosomal hub to shape both compartments and TADs.

In eukaryotes, nucleoli are the largest substructures of nuclei and participate in ribosome biogenesis[1]. In human cells, nucleoli assemble around hundreds of tandem repeat ribosomal DNA (rDNA) areas called nucleolar organizer regions (NORs), which are located on the short arms of acrocentric chromosomes 13, 14, 15, 21, and 22[2]. Nucleoli consist of three sub-regions: fibrillar centers (FCs), dense fibrillar components (DFCs) and granular components (GCs)[2]. In addition to its role in ribosome biogenesis, nucleoli also act as a central hub in regulating multiple nuclear processes necessary for maintaining cellular homeostasis, such as nuclear organization and energy metabolism[1,3,4]. Additionally, deficiencies in many of these processes have been linked to human diseases such as neurodegeneration and cancer[4].

[1]School of Life Sciences, Peking University, Beijing, China. [2]National Institute of Biological Sciences, Beijing, China. [3]State Key Laboratory of Membrane Biology, Biomedical Pioneering Innovation Center (BIOPIC), Peking University, Beijing, China. [4]Department of Biological Sciences, George Washington University Columbian College of Art and Sciences, Washington, DC, USA. [5]Institute of Health Service and Transfusion Medicine, Beijing, China. [6]National Biomedical Imaging Center, College of Future Technology, Peking University, Beijing, China. [7]State Key Laboratory of Proteomics, Institute of Basic Medical Sciences, National Center of Biomedical Analysis, Beijing, China. [8]Center for Statistical Science, Peking University, Beijing, China. [9]These authors contributed equally: Ting Peng, Yingping Hou, Haowei Meng. ✉e-mail: tingtingli@xmail.ncba.ac.cn; cheng_li@pku.edu.cn

Nucleoli-associated DNAs have been identified as nucleolus-associated domains (NADs) that are dispersed across all chromosomes, typically close to NORs, telomeres, and centromeres[5–7]. NADs are associated with repressive chromatin states, low gene density, and enrichment of repeated DNA sequences[5–7]. Moreover, the inactive chromosome X (Xi) is associated with nucleoli through the lncRNA Firre[8], suggesting a repressive regulation function of nucleoli; however, the organization of NADs around nucleoli remains unclear. Recently, chromosome conformation capture-derived approaches, especially Hi-C methods, have greatly promoted our understanding of hierarchical 3D genome structure in eukaryotic cells[9–11]. Chromosomes are mainly partitioned into compartments A and B, corresponding to euchromatin and heterochromatin, respectively[9], which are composed of smaller topologically associating domains (TADs)[12]. Accumulating evidence indicates that heterochromatin plays a key role in driving and maintaining genome architecture through a liquid-liquid phase separation (LLPS) mechanism[13–15]. Since nucleoli are the largest heterochromatin hubs in the nucleus, it is of great value to systematically study the features of nucleolus-associated chromatin interactions to uncover the role of nucleoli in 3D genome organization[12].

In the present study, we develop a variant of Hi-C experimental technique for enriching nucleolus-associated chromatin interactions, which combines nucleolus isolation with in situ Hi-C[10] experiments, and study the heterochromatin interactions within and around nucleoli in human cells at high resolution.

## Results

### Capturing nucleolus-associated chromatin interactions using nucleolus Hi-C

We developed a nucleolus Hi-C (nHi-C) protocol that applied in situ Hi-C[10] to isolated nucleoli, instead of nuclei, for the specific enrichment of nucleolus-associated chromatin interactions (Fig. 1a). nHi-C consists of typical in situ Hi-C steps, with the main difference being that the DNA digestion and proximity ligation procedures are performed using intact nucleoli instead of nuclei, enabling us to focus on chromatin interactions associated with nucleoli (Fig. 1a). The isolated nucleoli were evaluated by microscopy and further validated by western blotting of nucleoli marker proteins such as POLR1E and Nucleolin (Supplementary Fig. 1a, b). To better evaluate the effectiveness of nHi-C in enriching nucleolus-associated chromatin interactions, we simultaneously performed in situ Hi-C and nHi-C, in addition to whole genome sequencing (WGS) and nucleolus sequencing (NS)[16], to identify NADs in HeLa cells. Overall, we obtained a global chromatin interaction matrix with ~10 kb resolution from over 300 million effective read pairs for each Hi-C experiment (Supplementary Data 1). Nucleolus-associated DNA and whole genome DNA were sequenced at over 10x depth (Supplementary Data 2).

Comparison of the nHi-C matrix with the in situ Hi-C matrix demonstrates that interactions captured by nHi-C were enriched in certain genome regions (Fig. 1b–d) and in inter-chromosomal (trans) interactions (Fig. 1e, f). Both nHi-C and NS showed large variations in the read depth, while in situ Hi-C and WGS showed an almost uniform distribution (Fig. 1b, c, lower panel). In addition, nHi-C read depth signals resembled NS signals but not Hi-C signals, showing a genome-wide correlation of 0.7 with NS data but only a correlation of 0.4 with in situ Hi-C data (Supplementary Fig. 1c). NADs were identified by comparing NS data with WGS data or nHi-C data with in situ Hi-C data using two-state hidden Markov model (HMM) analysis[7]. nHi-C-identified NADs showed good consistency among three replicates, with greater than 90% overlap between nHi-C-identified NADs and NS-identified NADs (Supplementary Fig. 1d, e), demonstrating that nHi-C can enrich nucleolus-associated interactions. Consistent with previous findings that nucleolus-associated genome regions are mainly repressive heterochromatin[5–7,16,17], we found that more than 84% of nHi-C-identified NADs were located in

inactive B compartments (Supplementary Fig. 1f). Importantly, nHi-C increased the percentage of B-B compartment (B-B) interactions from 36.4% to 61.3% and trans B-B interactions from 3.8% to 6.7% as compared with in situ Hi-C (Fig. 1e, f, Supplementary Fig. 1g, h). These results not only demonstrate that nHi-C can stably enrich nucleolus-associated cis/trans interactions but also that most interactions captured by nHi-C are B-B interactions.

To confirm whether the distribution pattern of nucleolus-associated interactions is conserved across different cell types, we further performed nHi-C in the U2OS cell line (human bone osteosarcoma epithelial cells). The results showed that nHi-C increased the percentage of B-B interactions from 33.4% to 51.4% and trans B-B interactions from 5.1% to 8.3% (Supplementary Fig. 2a, b). Similar to the results in HeLa cells, nHi-C interaction-enriched regions in U2OS cells also showed high overlap with B compartments (Supplementary Fig. 2c), and the interactions captured by nHi-C were enriched in B-B interactions (Supplementary Fig. 2d).

### Validating hNADs by FISH in HeLa and U2OS cells

Since the NADs identified by HMM models were almost the same as B compartments (Supplementary Fig. 1f), consistent with previous findings that NADs cover about 40% of whole genome[7,17], we filtered the NADs using a stringent cutoff (read depth ratio of 2 between nHi-C and in situ Hi-C), and identified 264 regions consisting of 754.8 Mb chromatin regions or 52.6% NADs in HeLa cells, which we defined as high-confidence NADs (hNADs) (Fig. 2a, Supplementary Fig. 3a and Supplementary Data 3). Notably, hNADs showed significantly stronger inter-chromosomal B-B interactions and had on average 2.36-fold increased trans hNAD-hNAD interactions as compared with in situ Hi-C data (Supplementary Fig 3b, c). hNADs occurred in all chromosomes and accounted for chromosomal length ranging from 11.12% to 50.44%; however, smaller chromosomes tended to have a higher proportion of hNADs (Supplementary Fig. 3d, e), which is consistent with previous studies on NADs[5,6]. In addition, there was greater than 80% overlap between hNADs and the NADs identified by Nemeth et al. in HeLa cells using stringent cutoff parameters[5] (Supplementary Fig. 4a). To validate the localizations of hNADs in the nucleus, we selected seven representative hNADs loci from six chromosomes (Fig. 2b, Supplementary Fig. 4b–e) and performed high-resolution oligopaint DNA fluorescence in situ hybridization (FISH) experiments[18]. The results showed that both hNADs overlapped with previous study and hNADs exclusively reported in this study distributed tightly around the nucleolus (Fig. 2c–e, Supplementary Fig. 4f and Supplementary Movies 1–3), demonstrating the reliability of hNADs identified by nHi-C data.

Next, we compared hNADs between HeLa and U2OS cells. We identified 307 regions consisting of 635.9 Mb chromatin regions as hNADs in U2OS cells. There was an 94% overlap between hNADs in HeLa and U2OS cells (Supplementary Fig. 4g), indicating hNADs are conserved between different cell types, which is consistent with Nemeth et al. results in HeLa and IMR90 cells[5]. To validate the difference of hNADs between the two cell lines, two HeLa-specific hNADs and two U2OS-specific hNADs were selected and imaged by FISH experiments (Fig. 2b, f–k, Supplementary Fig. 4h, i and Supplementary Movies 4–5). It showed that HeLa-specific hNADs were adjacent to the nucleolus in HeLa cells, but far away from the nucleolus in U2OS cells (Fig. 2l–n, Supplementary Movies 6–7). On the contrary, U2OS-specific hNADs were close to the nucleolus in U2OS cells, but far away from the nucleolus in HeLa cells (Supplementary Fig. 4j–l, Supplementary Movies 8–9). These results confirm that nHi-C can identify cell-type specific NADs with high confidence.

### NOR-bearing chromosomes form two interacting clusters

Next, we focused on the trans interactions of the five NOR-bearing chromosomes. We first clustered all the chromosomes based on their trans interactions of hNADs using nHi-C and in situ Hi-C data. The

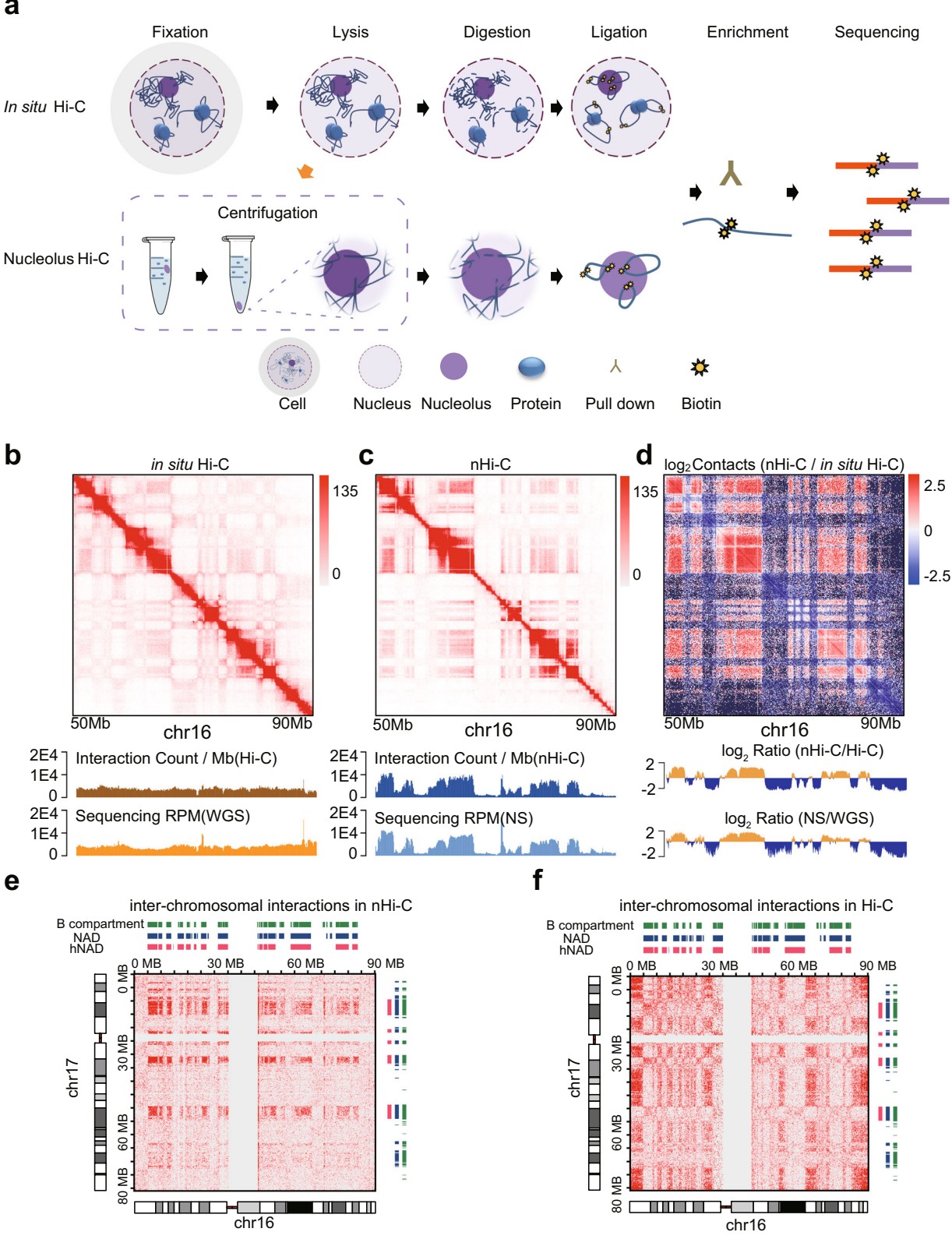

**Fig. 1 | nHi-C preferentially captures nucleolus-associated chromatin interactions. a** Flow chart of nHi-C. For comparison with in situ Hi-C, nHi-C was performed with isolated nucleoli. **b, c** Interaction heatmaps of in situ Hi-C and nHi-C. WGS: whole-genome sequencing, RPM: reads per million for each chromosome bin, NS: nucleolus sequencing. **d** Interaction heatmap showing the log2 fold-change in normalized interactions between nHi-C and Hi-C. NADs and nHi-C interaction-enriched regions are defined by an HMM model with log2Ratio (nHi-C/Hi-C) or log2Ratio (NS/WGS). **e, f** Interaction heatmaps between chromosomes 16 and 17 in nHi-C (**e**) and in situ Hi-C (**f**).

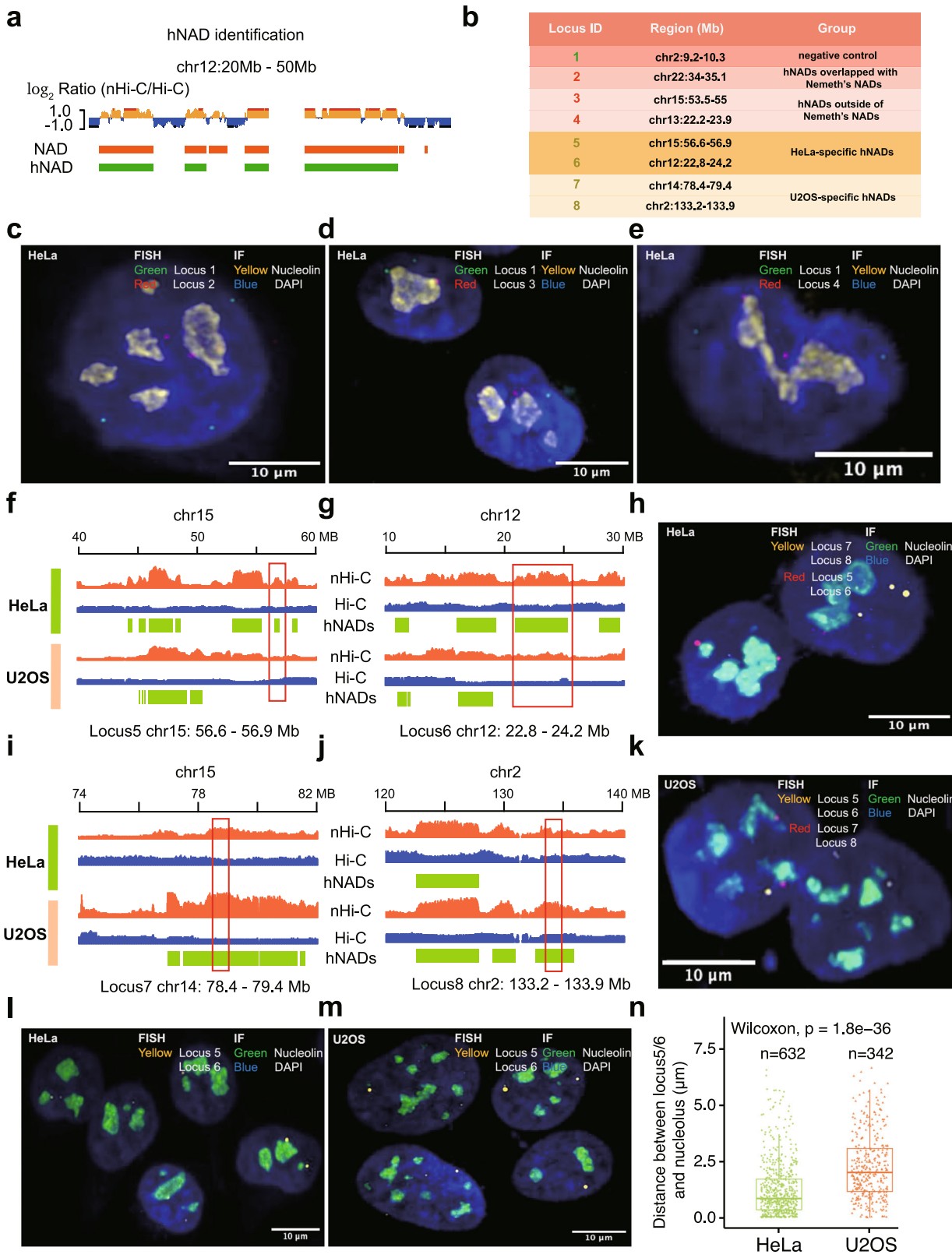

clustering results from the nHi-C data showed that three NOR-bearing chromosomes, 15, 21, and 22, formed the strongest *trans* interaction cluster, while the other two NOR-bearing chromosomes, 13 and 14, showed mild *trans* interactions that can hardly be distinguished from other chromosomes (Fig. 3a). The clustering results from the in situ Hi-C data did not reveal this phenomenon (Supplementary Fig. 5a). The different interaction pattern of these five NOR-bearing chromosomes

was further confirmed by detailed interaction heatmaps (Fig. 3b). Chromosomes 21 and 1 showed an unexpected high interaction, which was further confirmed as a translocation event involving these two chromosomes (Supplementary Fig. 5b). Chromosomes 13 and 14 showed significantly weaker interactions compared with the other three NOR-bearing chromosomes, and the signal ratio of hNADs in these two chromosomes was also much lower (Fig. 3c, d,

**Fig. 2 | hNADs in HeLa and U2OS cells. a** High-confidence NADs (hNADs) are defined as those NADs possessing a chromosomal bin with log$_2$Ratio greater than 1. **b** List of genome regions chosen to perform FISH experiments. **c–e** Representative images of hNADs' Oligopaint FISH probes (Locus 2-4) in HeLa cells. (*n* = 5 biologically independent samples). **f**, **g** Track plot of HeLa-specific hNADs chosen to perform DNA-FISH. **h** Representative images of cell type-specific hNADs' Oligopaint FISH probes (Locus 5-8) in HeLa cells (*n* = 5 biologically independent samples). **i**, **j** Track plot of U2OS-specific hNADs chosen to perform DNA-FISH. **k** Representative images of cell type-specific hNADs' Oligopaint FISH probes

(Locus 5-8) in U2OS cells (*n* = 3 biologically independent samples). **l**, **m** Representative images of HeLa-specific hNADs' Oligopaint FISH probes (Locus 5 and 6, the same probes were used for the two loci) in HeLa and U2OS cells (*n* = 5 biologically independent samples). **n** Distance between HeLa-specific hNADs (Locus 5 and 6) and nucleolus in HeLa and U2OS cells. In box-plots, center line stands for median; box limits are the 25th and 75th percentiles. '*n*' indicates the total number of cells imaged in independent experiments. Statistically significant differences are indicated and were calculated with two-sided Wilcoxon test.

Supplementary Fig. 5c). Heterochromatin histone modifications H3K9me3 and H3K27me3 were more enriched in chromosomes 15, 21, and 22, and the gene expression levels were also significantly lower in these three chromosomes (Supplementary Fig. 5d–g). Taken together, this evidence suggests that the five NOR-bearing chromosomes can be clustered into two groups: chromosomes 15, 21, and 22 that form strong hNAD *trans* interactions and are associated with more repressive chromatin states and weaker gene expression, and chromosomes 13 and 14 that form moderate *trans* interactions and are associated with higher gene expression levels. These results are consistent with those of a previous study showing that not all NORs can be transcribed into rRNA and inactive NORs localize outside the nucleolus[19]. Our results indicate that NORs on chromosomes 15, 21, and 22 may be more actively involved with nucleolus formation in HeLa cells.

### nHi-C enriches rDNA-associated chromatin interactions

Since nHi-C enriched nucleolus-associated interactions, we reasoned that nHi-C also can enrich rDNA-associated interactions. The chromatin interactions related to a specific rDNA repeat unit are difficult to study due to the highly repeated sequences of rDNA; thus, we treated all rDNA repeats as one unit and examined its chromatin interactions genome-wide, as was performed in a previous study using Hi-C datasets[20]. We found that nHi-C achieved a remarkable enrichment (7-fold) of rDNA-associated chromatin interactions as compared with in situ Hi-C results (Fig. 3e), demonstrating that nHi-C is a powerful tool for studying rDNA-related chromatin interactions. Specifically, approximately 76% of rDNA-associated interactions captured by nHi-C were within the rDNA repeat units (Fig. 3f). rDNA-interacting regions were significantly enriched in NADs (Supplementary Fig. 6a, b), and rDNA-interacting genes showed lower expression levels than those of other genes (Supplementary Fig. 6c). An rDNA repeat sequence is divided into 5' transcribed regions (TR, containing 18S, 5.8S, 28S) and the 3' non-transcribed intergenic spacer (IGS) region[21]. The *cis* interaction heatmap of the rDNA repeat region showed that TR and IGS formed separate topological domains (Fig. 3g). The IGS region contained two subdomains that strongly interact with each other, while the interaction was weak between the TR and IGS domains. In addition, NOR-bearing chromosomes did not display dramatically higher interaction frequencies with rDNA as compared with other chromosomes, suggesting spatial isolation of rDNA-containing NORs in the nucleoli from both NOR-bearing and non-NOR bearing chromosomes (Fig. 3h). In summary, nHi-C is able to enrich rDNA-associated interactions, which provides a high-resolution view of interactions within rDNA repeat units.

### Global hNAD *trans* interactions consist of three distinct clusters

Next, we investigated the *trans* interactions among the entire 264 hNADs. The hNADs were clustered into three subgroups: G1, G2, and G3, according to their *trans* interaction frequencies (Fig. 4a). G2 hNADs consist of 9% of the total hNADs and barely form any *trans* interactions, and they are mainly heterochromatins adjacent to centromeres with strong repressive epigenetic modifications H3K9me3 and H3K27me3 (Fig. 4b–e, Supplementary Fig. 7). G3 hNADs are frequently involved in forming *trans* interactions and consist of 25% of all hNADs, which are likely heterochromatins frequently surrounding nucleoli, such as large genomic regions of chromosomes 3, 6, 16, and 20 (Fig. 4b, h). We

further confirmed that G3 hNADs had significantly higher *cis* and *trans* interaction frequencies than the other two subgroups (Fig. 4f, g). G1 hNADs are heterochromatins far away from centromeres and form moderate interactions with G3 hNADs (Fig. 4c, i). A total of 61.3% of G1 hNADs and 40.1% of G3 hNADs overlap with lamina-associated domains (LADs), while only 25.2% of G2 hNADs overlap with LADs (Fig. 4b), which demonstrates the significant difference among the three hNAD subgroups for their tendency to switch between NAD and LAD after mitosis. We also noticed that some hNADs formed several hotspots of *trans* hNAD interactions, which were all validated as translocations rather than heterochromatin-interacting hubs (Supplementary Fig. 8).

### Nucleolus-associated *trans* interactions are enriched around centromeres

Quinodoz et al. reported that regions linearly close to the centromere are closer to the nucleolus[22]. Consistently, we found that hNADs were preferentially distributed close to centromeres (Fig. 4i), which was further confirmed by quantitative analysis (Supplementary Fig. 9a). In addition, the signal ratio of a hNAD is positively correlated to its proximity to centromere (Supplementary Fig. 9b). Consequently, we investigated whether *trans* interactions with NADs were also enriched around centromeres. Genome-wide analysis indicates that most hotspot regions that prefer to form *trans* interactions with NADs were distributed near centromeres (Fig. 5a). NAD-related *trans* interaction intensity was positively correlated with the proximity to centromeres based on the nHi-C data, while this pattern was not revealed by the in situ Hi-C data (Fig. 5b, c). For example, the *trans* interactions between chromosomes 2 and 6 were more significantly detected around centromeres in nHi-C data as compared with in situ Hi-C data (Fig. 5d, e). Therefore, nHi-C uncovered that nucleolus-associated *trans* interactions are enriched around centromeres.

### hNAD boundaries coincide with TAD boundaries and are demarcated by CTCF and repressive epigenetic histone marks

Since hNAD boundaries represent the border between the nucleoli periphery and the rest of the nuclear space, we investigated the features of chromatin marks and protein binding at hNAD boundaries. Systematic analysis of histone modifications shows that hNADs were mainly occupied by H3K9me3 and H3K27me3 modifications (Fig. 6a). hNAD boundaries sharply demarcated the chromatin with different histone modifications, where repressive histone modifications such as H3K27me3 and H3K9me3 were enriched within hNADs (Fig. 6b), and active histone modifications such as H3K4me3 and H3K36me3 were higher outside hNADs (Supplementary Fig. 10). By calculating the insulation score of each bin at 20 kb resolution from the in situ Hi-C data, we found that the insulation scores at hNAD and other NAD boundaries were significantly lower than those in surrounding regions, which is similar to the pattern of TAD boundaries (Fig. 6c), suggesting that NAD boundaries overlap with TAD boundaries. Moreover, the absolute value of insulation scores at the boundaries of hNADs were significantly higher than those at other NADs (Fig. 6c, d), which indicates that hNAD boundaries tend to overlap with strong TAD boundaries. Consistent with this, CTCF had stronger binding signals at hNAD boundaries (Fig. 6e). An example in Fig. 6f shows that TAD boundaries

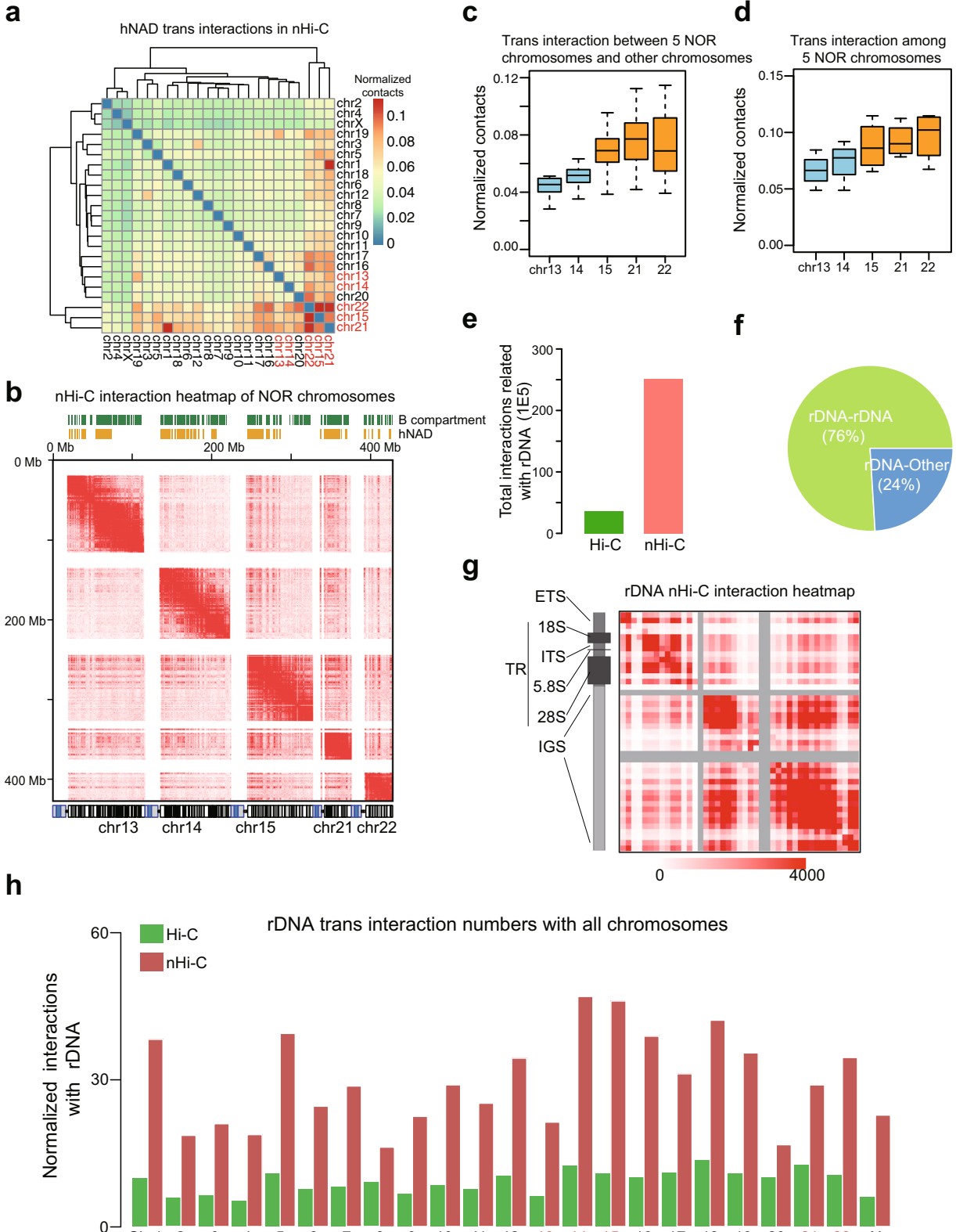

**Fig. 3 | Deciphering nucleolus-associated interactions using nHi-C.**
**a** Chromosome clustering results of hNAD *trans* interactions captured by nHi-C. The 5 NOR-bearing chromosomes were labeled in red. **b** Interaction heatmap of NOR-bearing chromosomes. The bottom panel represents cytoband staining of chromosomes: rDNA arms are colored in blue. **c** Box-plots (center line, median; box limits, the 25th and 75th percentiles) of *trans* interactions between the 5 NOR-bearing chromosomes and other chromosomes (*n* = 22 chromosomes). **d** Box-plots of *trans* interactions among the 5 NOR-bearing chromosomes (*n* = 4 chromosomes). **e** Number of rDNA-related interactions captured by in situ Hi-C and nHi-C. **f** Percentage of *cis* (rDNA-rDNA) and *trans* (rDNA-other genome regions) rDNA interactions in nHi-C. **g** *Cis* interaction heatmap of rDNA repeat units from nHi-C data. **h** The normalized number of rDNA-associated interactions in different chromosomes from in situ Hi-C and nHi-C data. The 5 NOR-bearing chromosomes were labeled in red.

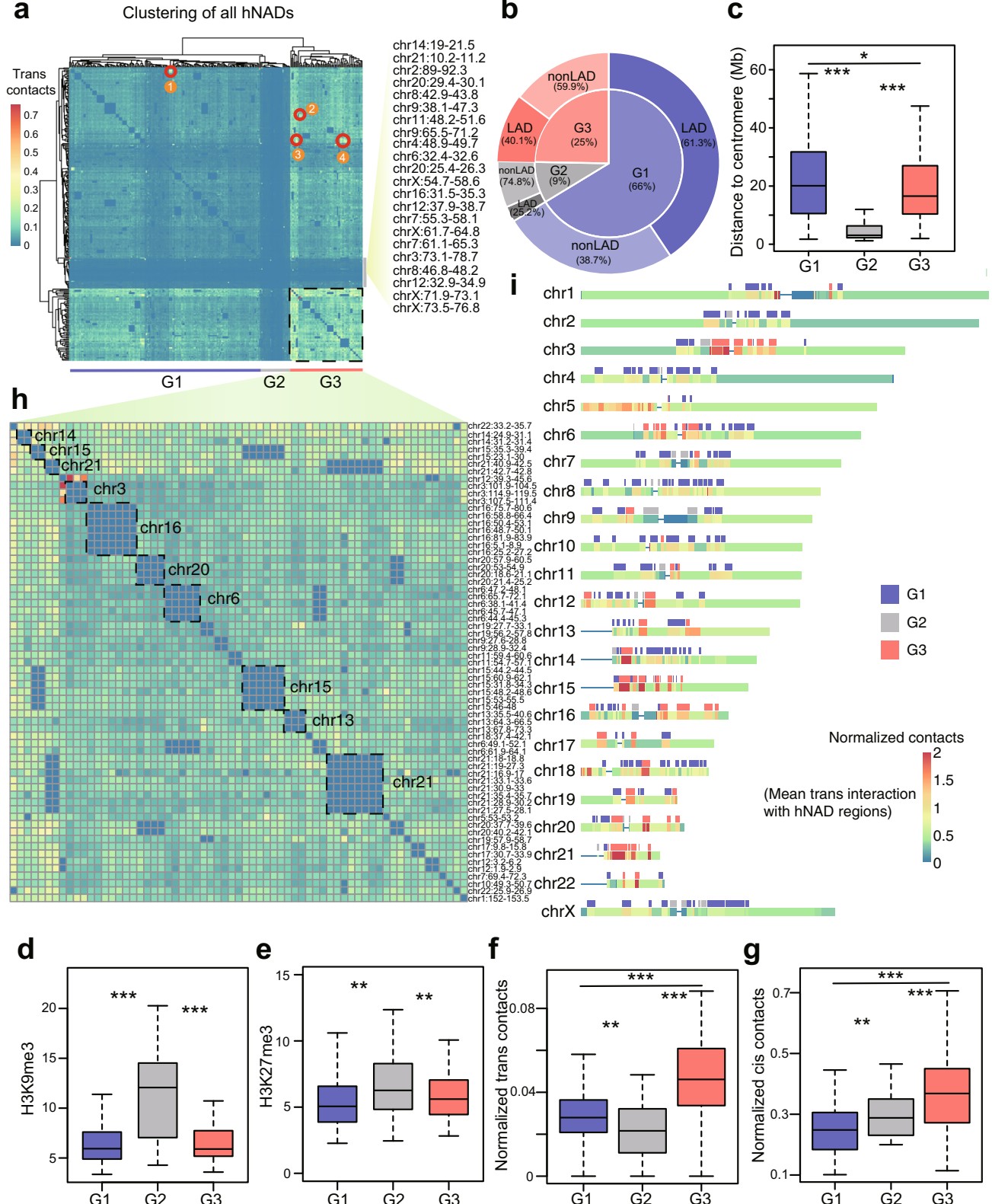

**Fig. 4 | Global properties of hNAD inter-chromosomal interactions. a** hNAD clustering results based on hNAD *trans* interactions captured by nHi-C. Circled out regions are hNAD-involved chromosome translocations. **b** Percentage of G1, G2, and G3 hNADs (inner layer) and their overlap percentage with LADs (outer layer). **c** Distance of G1, G2, and G3 hNADs from centromeres. **d**–**e** H3K9me3 (**d**) and H3K27me3 (**e**) signals in G1, G2, and G3 hNADs. **f**, **g** *Trans* (**f**) and *cis* (**g**) interaction levels of G1, G2, and G3 hNADs. **h** Composition and chromatin interactions of G3

hNADs. **i** Mean *trans* interaction level of chromosomal bins with other hNADs and the locations of G1, G2, and G3 hNADs. In all box-plots (**c**–**g**), center line stands for median; box limits are the 25th and 75th percentiles. $n = 177$ (G1), $n = 22$ (G2), $n = 65$ (G3). Statistically significant differences are indicated and were calculated with two-sided Wilcoxon test. $*p < 0.05$, $**p < 0.01$, $***p < 0.001$. Source data are provided as a Source Data file.

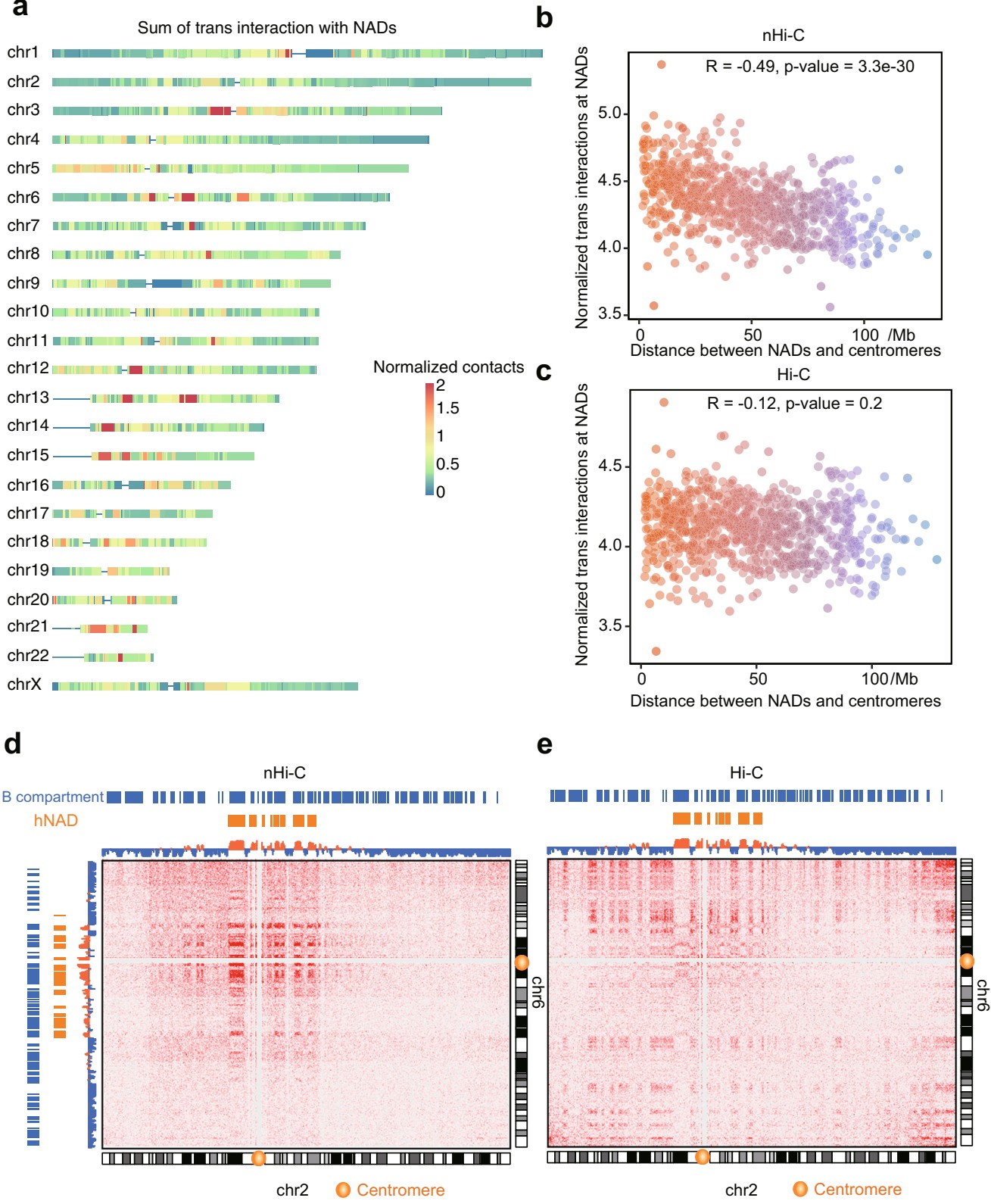

**Fig. 5 | Global distribution pattern of nucleolus-associated *trans* interactions. a** Sum of *trans* interactions with NAD regions for chromosomal bins. **b**, **c** Scatter plot of the number of *trans* interactions associated with NADs (*y* axis) and centromere distance (*x* axis) from nHi-C (**b**) and in situ Hi-C (**c**) data. Each point represents a chromosomal bin. Statistically significance was calculated with two-sided Pearson correlation test. **d**, **e** Interaction heatmap between chromosomes 2 and 6 from nHi-C (**d**) and in situ Hi-C (**e**) data. Source data are provided as a Source Data file.

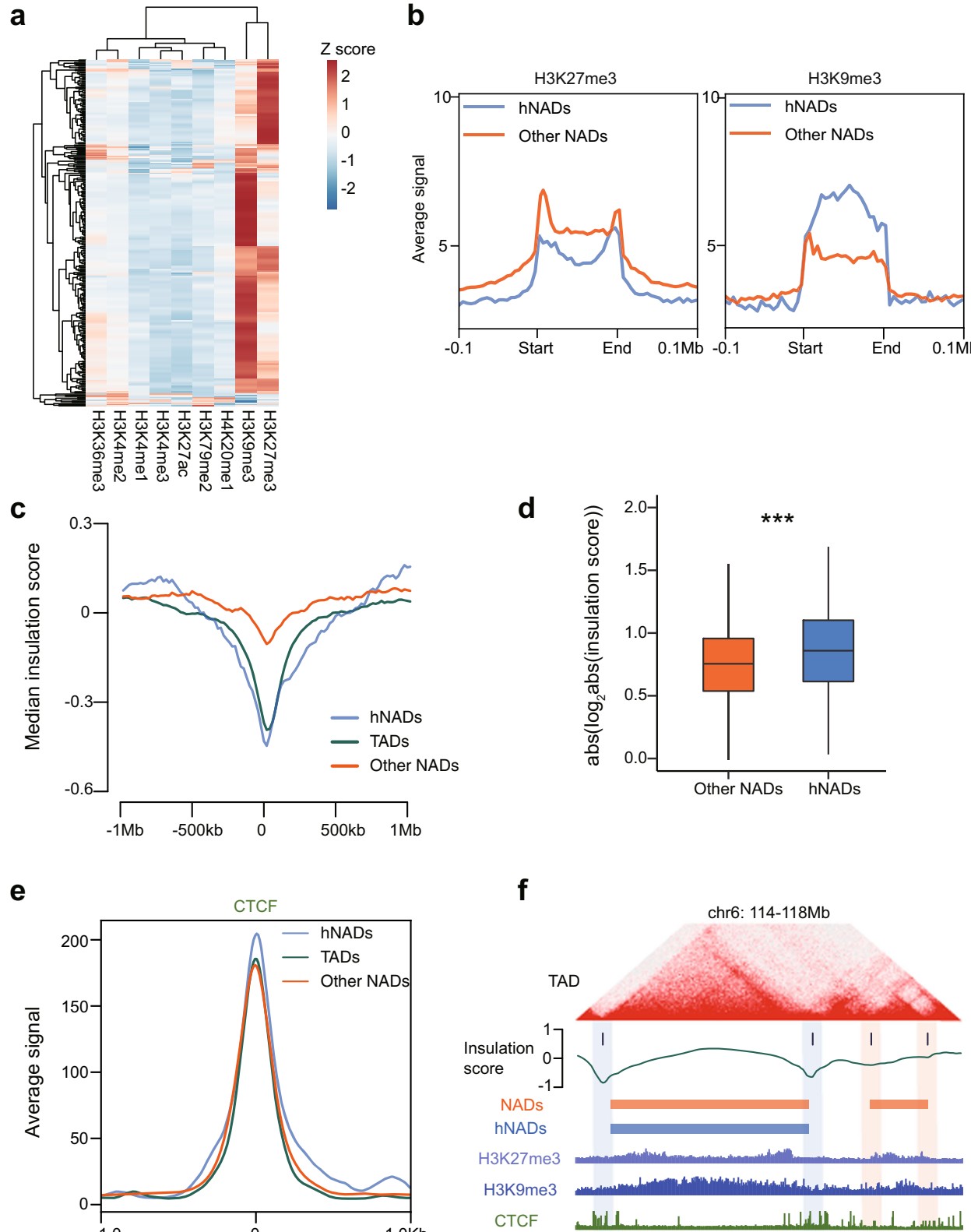

**Fig. 6 | Comparison of epigenetic signals between hNADs and TADs. a** Heatmap of histone modification signals (columns) in hNAD regions (rows). **b** Average H3K27me3 and H3K9me3 signals in hNAD, other NAD and surrounding regions. **c** Median insulation scores surrounding the boundaries of NADs and TADs. **d** Boxplots (center line, median; box limits, the 25th and 75th percentiles) of insulation scores at hNADs and other NADs. $n = 588$ TAD boundaries overlap with

hNAD boundaries, $n = 726$ TAD boundaries overlap with the boundaries of other NADs. Statistically significant differences were calculated with two-sided Wilcoxon test. ***$p < 0.001$. **e** CTCF binding signals at hNADs, other NADs, and TAD boundaries. **f** An example of overlapping NAD and TAD boundaries. Source data are provided as a Source Data file.

overlapping with hNADs are stronger than TAD boundaries overlapping with other NADs, and correspondingly have a higher CTCF binding signal. In addition, hNADs were frequently composed of several sequential TADs (Supplementary Fig. 10d). In summary, we found that hNAD boundaries coincide with TAD boundaries and hNADs are demarcated by repressive epigenetic histone marks.

## Genome structures reorganize in response to nucleolus disassembly

To reveal the causal influence of nucleoli on spatial genome structures, we performed nHi-C and in situ Hi-C experiments with HeLa cells treated with 10 ng/ml Actinomycin D (ActD), which causes the disassembly of nucleoli by inhibiting Pol I transcription activity[23]. Immunostaining of Nucleolin demonstrates the destructive effect of ActD on nucleoli, which makes nucleoli more fragile upon sonication and less efficient in isolating them (Supplementary Fig. 11, Supplementary Movies 10–11). nHi-C results reflected that the interaction frequencies inside hNADs were strongly down-regulated after ActD treatment (Supplementary Fig. 12a, b), resulting in a higher similarity between the nHi-C and Hi-C data of ActD-treated cells (Supplementary Fig. 12c). Results of the in situ Hi-C experiment showed that the ActD-treated sample displayed significantly increased *trans* interactions and decreased *cis* interactions (Fig. 7a), and the increased *trans* interactions were mainly *trans* B-B interactions, while the decreased *cis* interactions were mainly *cis* A-A interactions (Fig. 7b–d). In addition, we found that approximately 81.6% of the increased *trans* B-B interactions were related to hNADs (Fig. 7e), suggesting that changes in the *trans* interactions were mainly driven by hNADs.

While the changes in the *cis* hNAD-hNAD interactions caused by ActD were not related to the G1, G2, or G3 hNAD group, G3 hNADs had a significantly higher level of *trans* interactions than those in the G1 and G2 group (Supplementary Fig. 12d, e). We then analyzed the compartments and TADs following nucleolus disassembly, which revealed that the compartment strength was significantly decreased after ActD treatment, and the inter-compartmental interactions were increased and the A-A interactions were decreased (Fig. 7f, g). Similarly, there was an increased frequency of interactions across different TADs (Supplementary Fig. 12f, g). Specifically, we found that TADs located near hNAD boundaries showed a significantly higher number of interactions with their neighboring TADs after nucleolus disassembly (Fig. 7h, i), suggesting that the release of hNADs from dissociated nucleoli increased their interaction frequency with neighboring chromatin structures. In summary, these results show that nucleolus deformation has a large effect on genome structures and hNADs are hotspot regions of genome structural changes after nucleolus disassembly.

## Discussion

Nucleoli are the largest transcriptional machineries in eukaryotic cells, regulating key biological processes of ribosome biogenesis and organizing nuclear structures. The structure of nucleoli has been well studied, mainly using imaging technologies[24,25], but less commonly from the 3D genome perspective employing high-throughput sequencing. In the present study, we developed an nHi-C method and identified high-confidence nucleolus-associated chromatin interaction domains genome-wide.

Previous studies have demonstrated that not all NORs transcribe rRNAs and there is at least one non-competent NOR in HeLa cells[26], suggesting that although most non-competent NORs are associated with nucleoli, the five NOR-bearing chromosomes may have different chromatin organization patterns. These notions are consistent with our findings that chromosomes 15, 21, 22 show strong inter-chromosomal interactions, while chromosomes 13 and 14 show moderate interaction frequencies. The gene expression levels and epigenetic modifications among hNADs located on the five NOR-bearing chromosomes are also significantly different. We propose that the NORs on chromosomes 15,

21, and 22 prefer to be included in the same nucleolus as compared with those on chromosomes 13 and 14, leading to the transcription state differences of the NORs on these chromosomes.

We also investigated the chromosomal regions that frequently interact with rDNA genome-wide, since nHi-C significantly enriched rDNA-associated chromatin interactions as compared with Hi-C methods. Since actively transcribed rDNA is tightly surrounded by RNA polymerase aggregates inside the nucleolus, rDNA-related interactions should be lower at NORs involved in nucleolus formation. We noticed that chromosomes 15, 21, 22 did not show much difference in interacting with rDNAs compared to chromosomes 13 and 14, while chromosomes 14 and 15 showed the strongest interaction with rDNA.

The *cis* interaction heatmap of rDNA units indicates independent rDNA topological domains of transcribed regions and 3′ IGS regions. In light of the core-helix model for the topology of rRNA genes[27] and knowledge of higher-order rDNA architecture[28], we propose that the TR and IGS regions form separate spatial structures to facilitate rRNA transcription.

The results of our study also clarify the relationships between LADs and NADs. LADs show a highly dynamic spatial architecture and are reshuffled between the nuclear periphery and the inner location of the nucleus, especially the nucleoli, upon mitosis[29]. Global *trans* hNAD-hNAD interaction analysis identified three classes of hNADs: G2 hNADs are adjacent to centromeres, barely form inter-chromosomal interactions and rarely overlap with LADs; G3 hNADs have a higher interaction frequency with NOR-bearing chromosomes and show a moderate frequency of overlap with LADs; the remaining G1 hNADs are far away from centromeres and less frequently associated with nucleoli, and are likely the chromatin regions that prefer to switch between NADs to LADs after mitosis. In addition, the G1 and G3 hNADs are similar to the NAD/LAD subclass regions (NADs overlapping with LADs) defined by Ref. [30].

All hNADs are modified by repressive histone marks such as H3K9me3 and H3K27me3; however, we did not find a significant difference in epigenetic status between the G1 and G3 hNADs, demonstrating that epigenetic state is not the only key element that determines the interaction frequency between heterochromatin and nucleoli. Suppression of rDNA transcription by ActD significantly reduced the interactions within hNADs but increased the *trans* interactions among hNADs, demonstrating a key role of rDNA transcription or rRNA itself in determining the structure of nucleoli. Moreover, the signal ratio of hNADs is positively correlated with the proximity to centromeres. Based on these observations, we propose a model for the structural organization of nucleoli (Fig. 8): rRNAs transcribed from competent NORs serve as seeds to form phase-separated droplets around NOR regions by recruiting heterochromatins through RNAs and proteins, such as HP1α that recognizes H3K9me3[31], and centromeres are more easily attached to nucleoli due to their compact repressive epigenetic modifications. This model explains why inter-chromosomal interactions among hNADs are organized in a centromere-proximal manner.

In summary, we developed a nucleolus Hi-C method for studying nucleolus-associated chromatin interactions. We provide a global view of heterochromatin interactions distributed around nucleoli and demonstrate that nucleoli act as an inactive inter-chromosomal hub to shape both compartments and TADs.

## Methods
### Cell culture
HeLa and U2OS cells were cultured in DMEM medium (ATCC-30-2001) with 10% fetal bovine serum and 5% $CO_2$ at 37 °C. According to previous studies that ActD at low concentration (50 ng/ml) mainly affect RNAP I other than RNAP II (500 ng/ml)[32,33], we treated cells with low concentration ActD (10 ng /ml), which is proposed to specifically

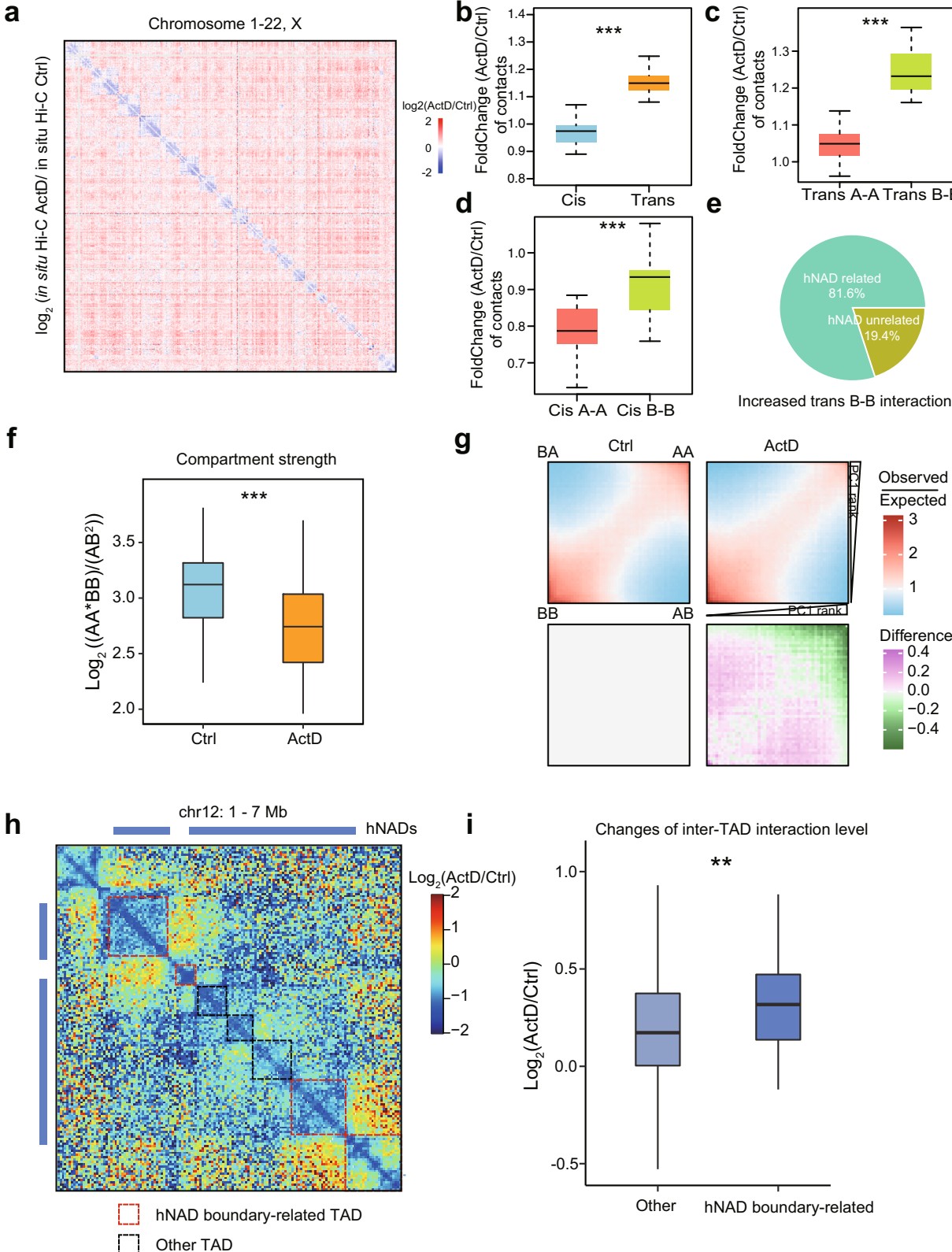

inhibit RNAP I. Then the cells were harvested to perform Hi-C and RNA-seq experiments.

## Hi-C and nHi-C experiments

Hi-C experiments were performed according to the in situ Hi-C protocol[10]. Briefly, about 1 million cells per sample were fixed with 1% formaldehyde and suspended in lysis buffer, then 100U DnpII restriction enzyme (NEB, R0147) was added. After overnight chromatin digestion, biotin was added to fill ends before DNA proximity ligation. Then DNA was sheared to short fragments using sonicator and 300–500 bp fragments were purified with T4 binding beads. Finally, biotin-labeled DNA was pulled down and libraries were amplified using

**Fig. 7 | Nucleolus disassembly results in genome reorganization. a** Interaction heatmap among all chromosomes showing the $\log_2$ fold-change in normalized interactions between ActD-treated (ActD) and untreated (Ctrl) HeLa cells. **b** Comparison of *cis* and *trans* interaction changes in all chromosomes after ActD treatment. **c** Comparison of *trans* A-A and *trans* B-B interaction changes in all chromosomes after ActD treatment. **d** Comparison of *cis* A-A and *cis* B-B interaction changes in all chromosomes after ActD treatment. **e** Most of the increased *trans* B-B interactions were related to hNADs. **f** Comparison of compartment strength in all chromosomes before and after ActD treatment. **g** Compartment saddle plot of A-A,

B-B, and A-B interactions before and after ActD treatment. **h** An example of changes in inter-TAD interactions around hNADs after ActD treatment. **i** Inter-TAD interactions were significantly increased at hNAD-related TADs. n = 324 TAD boundaries overlap with hNAD boundaries, $n = 528$ TAD boundaries overlap with the boundaries of other NADs. In all box-plots (**b–d**, **f**, **i**), center line stands for median; box limits are the 25th and 75th percentiles. Statistically significant differences were calculated with two-sided Wilcoxon test. $*p < 0.05$, $**p < 0.01$, $***p < 0.001$. In box-plots **b–d**, **f**, $n = 23$ chromosomes per group. Source data are provided as a Source Data file.

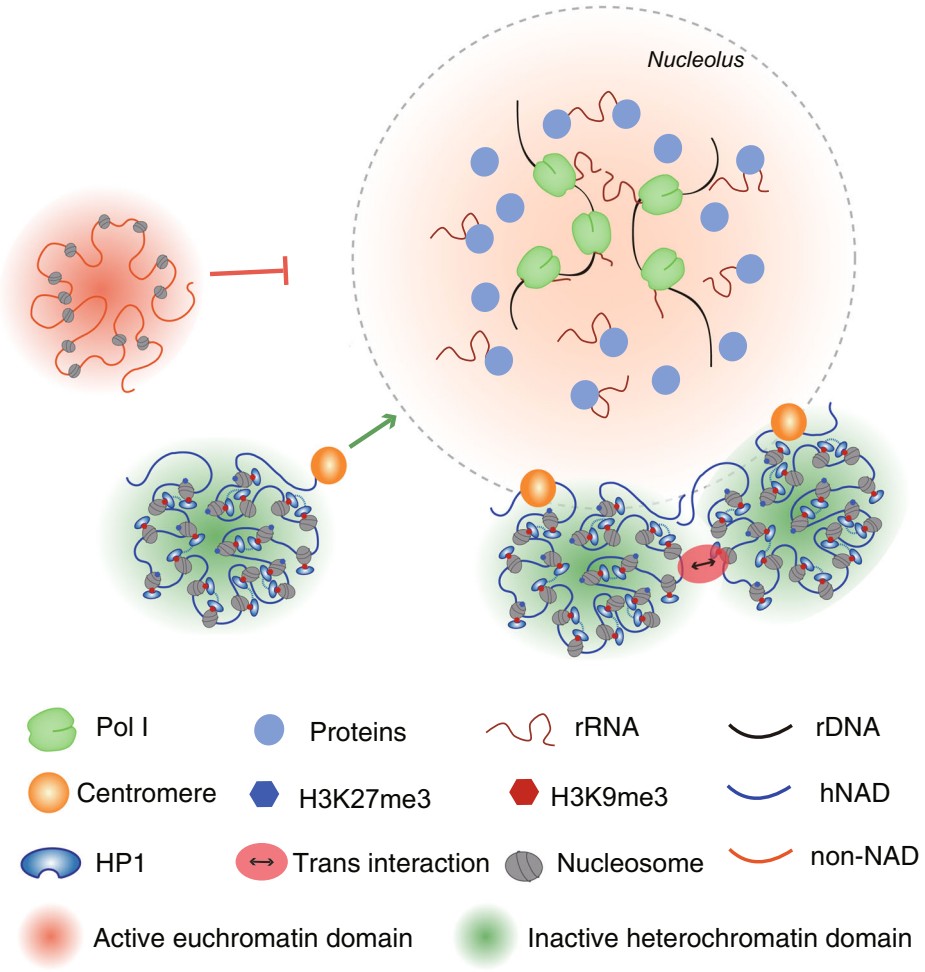

**Fig. 8 | A model of nucleolus-associated chromosome interactions.** rRNAs, as well as other proteins, aggregate around competent NORs by phase separation after mitosis, and the NOR phase serves as a signal source to recruit other

heterochromatin phases. Centromeres are more easily recruited to nucleoli due to their compact repressive epigenetic modifications; thus, hNAD inter-chromosomal interactions are organized around nucleoli in a centromere-proximal manner.

---

the NEBNext primer set. Nucleoli isolation were performed according to the protocol from previous study[34], and nHi-C experiments were modified as bellow.

1. Cell culture and crosslinking. Cells were grown to about 80% confluence under recommended conditions, harvested and suspended with PBS to a concentration of $1 \times 10^6$/ml maximum. Then the cells were cross-linked with formaldehyde at a final concentration of 1% for 10 min at room temperature (RT) with slow mixing. Glycine was added to a final concentration of 0.2 M to quench the reaction at RT for 5 min. Then cells were harvested and flash-frozen in liquid nitrogen and stored at −80 °C.

2. Nucleoli isolation. $2–5 \times 10^6$ cells were resuspended with 1.5 ml 0.5% sodium dodecyl sulfate (SDS) and incubate at 62 °C for 5–10 min in order to dissolve the cell membrane. Then 1.5 ml buffer S1 (0.25 M sucrose and 10 mM $MgCl_2$) were added, and the cells were broken with

ultra-sonication. The cells were monitored under a microscope until over 90% of them were burst, leaving intact nucleolus with cytoplasmic materials. Then the liquid was transferred to a 15 ml tube, and 3 ml buffer S2 (0.35 M sucrose, 0.5 mM $MgCl_2$) were carefully added to the bottom of the tube with the two layers cleanly separated. Centrifuge at $906 \times g$ for 5 min at 4 °C and discard the supernatant. The pellets were resuspended with 3 ml buffer S2, filtered with 5 um filter (Millipore) and transferred to a new 15 ml tube. 3 ml buffer S3 (0.88 M Sucrose, 0.5 mM $MgCl_2$) was added to the bottom of the tube with the two layers cleanly separated. The liquid was centrifuged at $1610 \times g$ for 10 min at 4 °C and the supernatant was discarded. Pure isolated nucleoli were resuspended with 100 ul buffer S3, checked under a microscope and stored at −80 °C.

3. Hi-C library construction. Nucleoli were collected by centrifugation at 300 g for 5 min, and resuspended with ice-cold Hi-C lysis

buffer (10 mM Tris-HCl pH 8.0, 10 mM NaCl, 0.2% Igepal CA630). The following endonuclease digestion, DNA purification, biotin ligation, and library construction steps are exactly following in situ Hi-C method.

## Immunoblotting

Protein extracts were run on 12% SDS-PAGE gels and transferred to nitrocellulose membranes with 0.45 μm pore size. Then membranes were probed with primary antibodies, Actin (ab8226, Abcam, dilution 1:1000), Nucleolin (ab129000, Abcam, dilution 1:100), Fibrillarin (ab4566, Abcam, dilution 1:100) and POLR1E (A12700, Abclonal, dilution 1:100) overnight at 4 °C. Membranes were washed with PBS, and incubated with a secondary antibody for 1 h at room temperature. For immunoblot visualization, ImageLab software was used.

## 3D DNA Fluorescence in situ hybridization coupled with immunofluorescence imaging

To label target genomic regions, Oligopaint FISH probes were designed using the OligoMiner[35] following methods described in a previous work[18]. Each target region was designed by a total number of 3000 DNA oligo probes. The primary oligopool was synthesized by Hongxun Biotech (Suzhou, China) and all the secondary probes were synthesized by Thermo Fisher Scientific Biotech (Shanghai, China). Primary oligo pools and secondary probe sets were shown in Supplementary Data 4 and 5, respectively.

3D DNA FISH-IF in HeLa and U2OS cell line was performed as previously described with minor modifications. Cells were grown on 35 mm glass bottom dish. After the coverage of cells reached 70–80%, cells were fixed using 4% PFA for 10 min at room temperature, washed with 1× PBS for three times, then permeabilized with 1% Triton X-100 in PBS for 15 min at room temperature and treated with 100 ug/ml RNase A at 37 °C for 45 min. The cells were rinsed once with 2× SSC, then incubated in 50% of formamide with 2× SSC overnight (12–16 h) at 4 °C. In the next day, the cells were pre-denatured at 78 °C for 10 min and rinsed by 70%, 85% and 100% cold ethanol sequentially. Simultaneously, 5 μL of the synthesized primary probes and 1 μL of the synthesized secondary probes (100 μM) were mixed in 35 μL 100% of formamide, shaken at 37 °C in a 1233 × g vortex for 15 min, and then added with 1 ul 100% of Triton X-100, 45 μL of 4× SSC/20% dextran sulfate for additional 30 min shaking. Next, the hybridization mix was denatured at 86 °C for 3 min and put immediately on ice. Then load the cells, and the samples were co-incubated at 86 °C for 3 min and put immediately on ice, finally hybridized at 37 °C overnight (16–20 h) in a dark and humidified chamber. In the 3rd day, the samples were post-hybridization washed with 2× SSC/1% Triton X-100 for three times at room temperature. Then washed with 2× SSC/1% Triton X-100 two times at 45 °C and 2xSSC/1% Triton X-100 two times at room temperature.

Immuno-flourescence is followed by FISH. Samples were permeabilized with 1% Triton in PBS for 10 min and then blocked in blocking buffer containing 5% BSA and 1% Triton in 1× PBS for 30 min. The samples were then incubated with primary antibody in blocking buffer for 2 h at room temperature, washed with 1× PBS three times, and then stained with secondary antibodies in blocking buffer for 1 h at room temperature. The labeled cells were washed again with 1× PBS, then post-fixed with 4% PFA for 10 min and finally stained with DAPI.

Primary antibody used in this study was nucleolin (ab129000, Abcam, dilution 1:100), Secondary antibodies were donkey anti-rabbit Alexa Fluor 488 (A-21206, Thermo Fisher Scientific, dilution 1:200) and goat anti-rabbit Cy3 (A-10522, Thermo Fisher Scientific, dilution 1:100).

OligoFISH-IF image acquisition was conducted with Live SR CSU W1 spinning disk confocal microscope (Nikon). The microscope

utilized a 100×, 1.40 NA oil-immersion objective lens (100×, 1.40 NA, Nikon TiE). For z stack imaging, 0.1μm/frame was set up for 3 um thickness in total. 3D FISH-IF imaging movies consisted of 31 frames for the 647-nm, 561-nm, 488-nm and 405-nm channels.

## OligoFISH-IF imaging analysis

The quantification of images was calculated by ImageJ (Fiji). Batch processing analysis of distance from FISH signal to the nearest nucleolus IF signal was performed using custom MATLAB code (MATLAB 2022b). Each frame of the nucleolus raw image was first binarized by Ostu algorithm. After removing the clusters less than 4 pixels, the binarized image was dilated and eroded by a 5 pixels' disk. Afterwards, the hole of binarized image was filled. The identified regions were defined as the nucleolus. For each frame of FISH raw images, the foreground was recognized by the comparison between raw image and mean filtering image. The intensity and area thresholds were used to remove non-specific signals. Clusters with more than three consecutive frames occurring in the z direction were retained, and centroid positions were calculated. The distance between FISH cluster and nucleolus was calculated as the closest distance between centroid position of FISH cluster and binarized nucleolus regions. The MATLAB code for tricolor and bicolor analysis are available on GitHub (https://github.com/ChengLiLab/nHi-C).

## DNA sequencing data processing

We prepared HeLa cells and then obtained the HeLa cells whole genome sequencing (WGS) data, HeLa nucleolus sequencing (NS) data with 150 bp paired-end sequencing on Illumina XTen platform. We processed these DNA sequencing data using the normal pipeline: bowtie2 v2.3.5 mapping, samtools v1.9 sorting, and removing duplicates with Picard v2.21.3. To ensure the mapping quality, especially in the NS data set, we only kept the mapped reads with a MAPQ value higher than 30.

## In situ Hi-C and nHi-C data processing

For all Hi-C data sets, we used the HiC-Pro[36] v2.11.12 analysis pipeline. We also added the rDNA 43 kb repeat unit sequence as an artificial chromosome to the hg19 reference genome. We only kept the valid interaction pairs with MAPQ higher than 30. We used the ICE method to normalize the interaction matrix for all in situ Hi-C matrixes.

For compartment and TAD analysis, compartment was assigned based on the PC1 score after PCA analysis of ICE normalized 100 kb matrix. TAD was called by GENOVA[37] package. Compartment saddle plot and TAD ATA (aggregate TAD analysis) analysis were also performed by GENOVA.

## Identification of NADs and hNADs

After mapping the DNA and Hi-C sequencing data to the reference genome, we calculated the sequencing depth (SD) and enrichment ratio (ER) as $ER = \log_2(NS\ SD\ /\ WGS\ SD)$ or $ER = \log_2(nHi\text{-}C\ SD/in\ situ\ Hi\text{-}C\ SD)$ at each genome position in each data set. And then based on genome sequencing ER, we used two-state hidden Markov model (HMM)-based analyses to assign regions as either NADs or inter-NADs as in Dillinger et al[7]. Similar analysis method was used to identify Hi-C NADs based on ER of Hi-C data. NADs containing more than one chromosomal bin with ER higher than 1 were defined as hNADs.

## Identification of chromosome translocations

Translocations were identified by HiNT[38] with 1 Mb and 100 kb in situ Hi-C matrix.

## RNA sequencing data processing

The RNA sequencing data was aligned by STAR[39] v2.7.2b and reads were counted by HTSeq v2.0.2.

## Statistically significant test

We used hypergeometric distribution to compute *P* values, testing whether two sets significantly overlapped. Statistical analysis was also performed by two-sided Wilcoxon test and Pearson correlation test. Significance was represented by *$p < 0.05$; **$p < 0.01$; ***$p < 0.001$.

## Reporting summary

Further information on research design is available in the Nature Portfolio Reporting Summary linked to this article.

## Data availability

The data that support this study are available from the corresponding authors upon reasonable request. The sequencing data generated in this study have been deposited in the Gene Expression Omnibus (GEO) database under accession code GSE90003. The histone modification and CTCF ChIP-Seq data were downloaded from ENCODE. The ENCODE IDs are: ENCFF000BAJ [https://www.encodeproject.org/experiments/ENCSR000AOA/], ENCFF000BCO [https://www.encodeproject.org/experiments/ENCSR000AOF/], ENCFF000BBG [https://www.encodeproject.org/experiments/ENCSR000AQO/], ENCFF000BBS [https://www.encodeproject.org/experiments/ENCSR000APB/], ENCFF000BCA [https://www.encodeproject.org/experiments/ENCSR000AOD/]. Source data are provided with this paper.

## Code availability

All essential codes of data analysis and figures for reproducible research are available on GitHub (https://github.com/ChengLiLab/nHi-C; https://doi.org/10.5281/zenodo.7466530[40]).

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

## Acknowledgements

This work was supported by National Natural Science Foundation of China (32288102, 32025006, 31871266) and National Key Research and Development Program of China (2021YFA1100300). Part of the data analysis was performed on the High Performance Computing Platform of the Center for Life Sciences, Peking University. We thank Guoliang Li and Yang Chen for critical comments.

## Author contributions

T.L. and C.L. designed the study. T.L., Y.H. and Y.C. performed main wet lab experiments, L.J. and Q.C. participated in the experiments. T.P., H.M. and T.L. analyzed the data and prepared the figures. X.W. analyzed FISH data. Y.S. supervised the FISH imaging validation. Y.Z. made Fig. 8. T.L., T.P., H.M., H.C. and C.L. wrote the manuscript.

## Competing interests

The authors declare no competing interests.
