## [Peer Review File · Nature Communications]

REVIEWER COMMENTS

Reviewer #1 (Remarks to the Author):

The authors present a novel approach to analysis of an important genomic feature, the subset of the genome localized around the nucleolar periphery. Multiple approaches have been used to map such Nucleolus-Associated Domains ("NADs"). These include classical biochemical isolation of nucleoli (van Koningsbruggen S, et al. *Mol Biol Cell* (2010); Nemeth et al. *PLoS Genetics* (2010); Dillinger et al. *PLoS One* 2017; Vertii et al. *Genome Res.* (2019); Bizhanova et al. *Chromosoma* (2020); Lu et al. *Cell Rep.* 30, 3296-3311 (2020)), FACS-mediated sorting of tagged nucleoli from cell extracts (Pontvianne et al. *Cell Rep.* (2016)), Dam-ID-based methods using either nucleolus-targeting peptides (Wang et al. *Genome Biology*, 14 Jan 2021, 22(1):36) or histone-peptide fusions (Bersalglieri et al. *Nat. Commun.* 13(1):1483, 18 Mar 2022), and most importantly in comparison to the present work, computational analysis of Hi-C datasets to pullout rDNA-containing junction pairs (Yu and Lemos, *PLoS Genetics* (2018)). Here, the authors combined biochemical purification with Hi-C analysis of the purified nucleoli ("nHi-C"). A strong aspect of this paper is that the authors compare side-by-side data from nHi-C with a traditional biochemical approach. These data suggest that nHi-C is a good way to map NADs (e.g. Fig. 1B-D), with the additional benefit of providing novel data regarding pairwise interactions. Therefore, the manuscript could be of high impact to an important field.

However, given the extensive previous work in the field listed above, much more needs to be done here to validate the current method, and to compare it to previous methods. For example, previous studies have used rDNA junction fragments from Hi-C datasets to analyze NADs (see the paper from the Lemos group listed above, and their other work on this subject). It is clear that purification of nucleoli prior to the Hi-C steps enriches for rDNA junctions (Figure 2H), but the authors need to test whether that changes the type of nucleolar trans interactions detected, or just changes their abundance in the library. Also, it is clear that the authors' set of hNADs in HeLa cells are much more extensive than those described in a previous paper (Fig. S3E, Dillinger et al. *PLoS One* 2017). A new technique like this requires validation via an orthogonal assay. Therefore, the authors need to perform DNA-FISH studies to determine whether hNAD peaks that were outside of the previous published data are indeed associated with nucleoli in cells. Furthermore, there are several comparisons that are left out of the paper. Where are the NAD maps from U2OS cells and how do they differ from HeLa? Can the authors show that differences in NADs between HeLa and U2OS detected via nHi-C can be validated by FISH? Likewise, why is there no map of NADs from the ActD treatment experiments? The latter figures would help shape the model in Fig. 7.

One other technical point- since ActD is so globally disruptive to nucleolar structure, the biochemical validation steps done in Fig. S1 should be applied to those preparations as well.

Regarding the model, it suggests that rDNA transcription occurs in a nucleolar region exterior to ribosome assembly. That is not how nucleolar architecture is usually viewed and should be corrected.

Reviewer #2 (Remarks to the Author):

In the manuscript "Mapping Nucleolus-associated Chromatin Interactions Using Nucleolus Hi C Reveals Pattern of Heterochromatin Interactions" authors developed nucleolus Hi-C (nHi-C) experimental technique that is aimed to enrich nucleolus-associated chromatin

interactions and combines nucleolus isolation with Hi-C in situ approach. The developed technique was employed to study the heterochromatin interactions around human nucleoli at high genomic resolution. nHi-C can stably enrich nucleolus-associated interactions with most interactions captured by nHi-C being B-B interactions.

Authors represent high quality original data that is clearly summarized in 7 main and 10 supplementary figures. The manuscript is logically written and the data for the most part are convincing. However the following questions should be addressed to confirm the conclusions drawn:

Questions:

Q1. Can we verify the nHi-C data by analyzing nucleolus-associated genome regions in Hi-C in situ data? What are the advantages of employing nHi-C technique in comparison to analyzing nucleolus-associated genome regions in Hi-C in situ experiment?

Q2. The data obtained also confirmed the well-established from 3D-FISH results fact that nucleolus-associated interactions are enriched around centromeres. Please add the comparison of the obtained nHi-C data with previous 3D-FISH results to the discussion.

Q3. It should be noted that rDNA-related interactions should be lower in NORs involved into nucleolus formation since inside the nucleolus rDNA is tightly surrounded by RNP aggregates. rDNA-related interactions could thus demonstrate different profiles in active and inactive NORs within the same cell line. Please add a discussion in this paragraph.

Major comments:

C1. Lines 249-251 – Not only RNA polymerase I but also RNA polymerase II is sensitive to Actinomycin D treatment. Please indicate how these effects were separated and provide more details in the methods.

C2. Please provide additional evidence and references that centrifugation is enough to prepare nucleolus fraction.

C3. It would be beneficial to confirm hNAD identification and localization by 3D-FISH.

C4. Introduction lines 40-53 – By my opinion, this part of the text could be shortened because it describes the basic concepts of the nucleolus known from textbooks.

Minor comments:

C1. Lines 155-162 – Please indicate which NORs are active and involved into nucleolus formation in the analyzed cell lines.

Point-to-point Response to Reviewer #1

The authors present a novel approach to analysis of an important genomic feature, the subset of the genome localized around the nucleolar periphery. Multiple approaches have been used to map such Nucleolus-Associated Domains (“NADs”). These include classical biochemical isolation of nucleoli (van Koningsbruggen S, et al. *Mol Biol Cell* (2010); Nemeth et al. *PLoS Genetics* (2010); Dillinger et al. *PLoS One* 2017; Vertii et al. *Genome Res.* (2019); Bizhanova et al. *Chromosoma* (2020); Lu et al. *Cell Rep.* 30, 3296-3311 (2020)), FACS-mediated sorting of tagged nucleoli from cell extracts (Pontvianne et al. *Cell Rep.* (2016)), Dam-ID-based methods using either nucleolus-targeting peptides (Wang et al. *Genome Biology*, 14 Jan 2021, 22(1):36) or histone-peptide fusions (Bersalghieri et al. *Nat. Commun.* 13(1):1483, 18 Mar 2022), and most importantly in comparison to the present work, computational analysis of Hi-C datasets to pullout rDNA-containing junction pairs (Yu and Lemos, *PLoS Genetics* (2018)). Here, the authors combined biochemical purification with Hi-C analysis of the purified nucleoli (“nHi-C”). A strong aspect of this paper is that the authors compare side-by-side data from nHi-C with a traditional biochemical approach. These data suggest that nHi-C is a good way to map NADs (e.g. Fig. 1B-D), with the additional benefit of providing novel data regarding pairwise interactions. Therefore, the manuscript could be of high impact to an important field.

Response: Thanks for the reviewer’s recognition of our work.

1. It is clear that purification of nucleoli prior to the Hi-C steps enriches for rDNA junctions (Figure 2H), but the authors need to test whether that changes the type of nucleolar trans interactions detected, or just changes their abundance in the library.

Response: Thanks for the reviewer’s insightful suggestions. To evaluate whether nHi-C change the type of nucleolar trans-interactions detected, we compared the trans interactions among hNADs captured by nHi-C and Hi-C. It showed that most hNADs trans interactions have much weaker signals in Hi-C compared to nHi-C, while hNADs

trans-interaction frequencies in nHi-C data were positively correlated with the results in Hi-C data (Fig. R1A).

More importantly, we concluded that the type of nucleolar trans interactions cannot be properly detected by Hi-C methods. As in the main text shows, ActD treatment which destroys nucleolar structures has no significant effect on cis hNAD-hNAD interactions (main text, Fig. S12D) and very weak effect on trans hNAD-hNAD interactions (main text, Fig. S12E). This was further confirmed by global hNAD-related interactions with or without ActD treatment (Fig. R1B-I). The nucleolus-related chromatin interactions were mixed up with high background genome-wide interactions that can hardly indicate nucleolar structures (Fig. R1D-I).

Combined with the previous data (Fig. R1D,G), we concluded that nHi-C experiments significantly enrich nucleolar trans-interactions while mainly remaining the type of nucleolar trans-interactions unchanged, and Hi-C method is not proper to study nucleolar-related chromatin interactions.

Fig. R1 (A) Scatter plot of hNADs trans-interaction frequency in nHi-C and Hi-C results. **(B)** Scatter plot of hNADs trans-interaction frequency before and after ActD treatment in Hi-C results. **(C)** Percentage of hNADs trans interaction detected before (Ctrl) and after (ActD) ActD treatment in Hi-C results. **(D-F)** Interaction heatmaps between chromosomes 16 and 17 in nHi-C **(D)**, *in situ* Hi-C **(E)** and *in situ* Hi-C after ActD treatment **(F)**. **(G-I)** hNADs interaction heatmaps between chromosomes 16 and 17 in nHi-C, **(G)** *in situ* Hi-C **(H)** and *in situ* Hi-C after ActD treatment **(I)**.

2. It is clear that the authors' set of hNADs in HeLa cells are much more extensive than those described in a previous paper (Fig. S3E, Dillinger et al. PLoS One 2017). A new technique like this requires validation via an orthogonal assay. Therefore, the authors need to perform DNA-FISH studies to determine whether hNAD peaks that were outside of the previous published.

Response: Thanks for the reviewer's valuable suggestions. To validate the hNADs we identified, we selected three representative hNADs (one shared hNAD between this study and Dillinger et al. PLoS One 2017, and two exclusively reported hNADs in our work, Table R1, Fig. R2A-D) and one non-hNADs as negative control to perform DNA-FISH experiments. The results showed that both shared NADs and NADs exclusively reported in this work distributed in close proximity to the nucleolus (Fig. R2E-G), demonstrating the reliability of hNADs identified by nHi-C data. We added the figures in Fig. 2 and Fig. S4, and updated the manuscript in page 7 lines 146-152.

Locus ID	Region (Mb)	Group
1	chr2:9.2-10.3	negative control
2	chr22:34-35.1	hNADs overlapped with previous study
3	chr15:53.5-55	hNADs outside of previous study
4	chr13:22.2-23.9	hNADs outside of previous study

Table R1 FISH target regions

Fig. R2 (A-D) Track plot of genome regions chosen to perform DNA-FISH. **(E-G)** Representative images of hNADs' Oligopaint FISH probes in HeLa cells. (3D imaging results are provided as Supplementary Videos)

3. Where are the NAD maps from U2OS cells and how do they differ from HeLa? Can the authors show that differences in NADs between HeLa and U2OS detected via nHi-C can be validated by FISH?

Response: Thanks for the reviewer's valuable suggestions. It is important to explore the consistency of NADs in different cell lines. To check this point out, we identified the hNADs in U2OS cells and compared them with hNADs in HeLa cells. It showed that over 94% hNADs regions are overlapped between HeLa and U2OS cells, indicating hNADs are highly conserved between different cell types, which is consistent with Nemeth et al.'s results in HeLa and IMR90 cells (Nemeth et al., 2010). Only 67.9 Mb HeLa-specific and 28.4 Mb U2OS-specific hNADs were identified (Fig. R3A). To

validate the difference of hNADs between the two cell lines, two HeLa-specific hNADs and two U2OS-specific hNADs were selected and imaged by FISH experiments (Fig. R3B-D, F-G). It showed that HeLa-specific hNADs were tightly adjacent to the nucleolus in HeLa cells, but far away from the nucleolus in U2OS cells (Fig. R3E, I-K). On the contrary, U2OS-specific hNADs were tightly close to the nucleolus in U2OS cells, but far away from the nucleolus in HeLa cells (Fig. R3H, L-N). We added the figures in Fig. 2 and Fig. S4, and updated the manuscript in page 8 lines 153-164.

Fig. R3 (A) Overlap between hNADs in HeLa cells and U2OS cells. **(B)** List of cell type-specific hNADs chosen to perform DNA-FISH. **(C-D)** Track plot of HeLa-specific hNADs chosen to perform DNA-FISH. **(E)** Representative images of cell type-specific hNADs' Oligopaint FISH probes (Locus 5-8) in HeLa cells. **(F-G)** Track plot of U2OS-specific hNADs chosen to perform DNA-FISH. **(H)** Representative images of cell type-specific hNADs' Oligopaint FISH probes (Locus 5-8) in U2OS cells. **(I-J)** Representative images of HeLa-specific hNADs Oligopaint FISH probes (Locus 5 and 6) in HeLa and U2OS cells. **(K)** Distance between HeLa-specific hNADs (Locus 5 and 6) and nucleolus in HeLa and U2OS cells. **(L-M)** Representative images of U2OS-specific hNADs Oligopaint FISH probes (Locus 7 and 8) in HeLa and U2OS cells. **(N)** Distance between U2OS-specific hNADs (Locus 7 and 8) and nucleolus in HeLa and U2OS cells. (3D imaging results are provided as Supplementary Videos)

4. Likewise, why is there no map of NADs from the ActD treatment experiments? The latter figures would help shape the model in Fig. 7.

Response: Thanks for the reviewer's valuable suggestions. In fact, we had compared the nHi-C and Hi-C data after ActD treatment, but it didn't show evident enrichment at certain genome regions and the read depth distribution is similar between nHi-C and Hi-C data after ActD treatment (Fig. R4A and R4B). As you suggested below (Comment #5), we have added the biochemical validation experiments after ActD treatment (Fig. R5). The results indicated that ActD treatment with low concentration (10 ng/ml) strongly destroyed nucleolar structures and make nucleoli more fragile upon sonication, resulting in insufficient isolation of nucleoli with gradient centrifuge. Thus, the nHi-C data with ActD treatment is pretty much the same as Hi-C data.

Fig. R4 (A) Correlation of read depth in Hi-C and nHi-C data. **(B)** Reads coverage plot of chr16.

5. One other technical point- since ActD is so globally disruptive to nucleolar structure, the biochemical validation steps done in Fig. S1 should be applied to those preparations as well.

Response: Thanks for the reviewer's suggestions. We added the biochemical validation experiments in HeLa cells after ActD treatment. The results indicated that ActD treatment with low concentration (10 ng/ml) strongly shrink nucleolar structures and make nucleoli more fragile upon sonication, resulting in bad isolation of nucleoli with gradient centrifuge (Fig. 5R). We added the validation results in Figure S11.

Fig. R5 (A) Isolated nucleoli after ActD treatment under microscopy. **(B)** Western blotting of nucleolar proteins from isolated nucleoli after ActD treatment. **(C)** Immunostaining results of nucleolin in ActD non-treat (left) and ActD treated HeLa cells (right). (3D imaging results are provided as Supplementary Videos)

6. Regarding the model, it suggests that rDNA transcription occurs in a nucleolar region exterior to ribosome assembly. That is not how nucleolar architecture is usually viewed and should be corrected.

Response: Thanks for the reviewer's valuable suggestions. We have revised our model and moved rDNA to the center of nucleolus (Fig. R6). (Page 39 line 604 of manuscript)

Fig. R6 A revised model of nucleolus-associated chromosome interactions.

Point-to-point Response to Reviewer #2

In the manuscript “Mapping Nucleolus-associated Chromatin Interactions Using Nucleolus Hi-C Reveals Pattern of Heterochromatin Interactions” authors developed nucleolus Hi-C (nHi-C) experimental technique that is aimed to enrich nucleolus-associated chromatin interactions and combines nucleolus isolation with Hi-C in situ approach. The developed technique was employed to study the heterochromatin interactions around human nucleoli at high genomic resolution. nHi-C can stably enrich nucleolus-associated interactions with most interactions captured by nHi-C being B-B interactions. Authors represent high quality original data that is clearly summarized in 7 main and 10 supplementary figures. The manuscript is logically written and the data for the most part are convincing.

Response: We thank the reviewer for the positive comments and constructive suggestions.

Q1. Can we verify the nHi-C data by analyzing nucleolus-associated genome regions in Hi-C in situ data? What are the advantages of employing nHi-C technique in comparison to analyzing nucleolus-associated genome regions in Hi-C in situ experiment?

Response: Thanks for the reviewer’s valuable and critical questions. Both nHi-C and Hi-C can capture nucleolus-associated *cis* interactions, while nHi-C highly enriched NADs related heterochromatin contacts (Fig.R7A-B). Therefore, with same sequencing depth, nHi-C can provide a higher resolution of nucleolus-associated *cis* interactions than Hi-C. In addition, nHi-C also showed evident enrichment of *trans* interactions (Fig.R7C-D, F-G). Based on the high resolution *trans* interaction heatmap, nucleolus-specific trans-interaction patterns were uncovered, including that the NOR-bearing chromosomes formed into two clusters and hNADs clustered into three groups

in HeLa cells. Since Hi-C has a bias to capture interactions formed in active euchromatin regions, it cannot indicate the pattern of *trans* nucleolus-associated interactions correctly.

Besides the significant enrichment of nucleolus-associated cis/trans interactions of nHi-C over traditional Hi-C methods, we propose that nHi-C is the proper method to study nucleolus-associated genome interactions other than Hi-C methods, as Hi-C cannot truly reflect the nucleolar cis/trans interactions. We have showed that ActD treatment has no significant effect on cis hNAD-hNAD interactions (main text, Fig. S12D) and very weak effect on trans hNAD-hNAD interactions (main text, Fig. S12E). As Hi-C method globally captures chromatin interactions from millions of cells together, it captures the interactions of nucleolus-associated genome regions with no difference even if the nucleolar structures are destroyed with ActD treatment (Fig. R7E, H). The nucleolus-related chromatin interactions were mixed up with high background genome-wide interactions that can hardly indicate nucleolar structures (Fig. R7D,E,I,G), we concluded that nHi-C experiments significantly enrich nucleolar cis/trans-interactions and reveal specific pattern of *trans* nucleolus-associated interactions, and Hi-C method is not proper to study nucleolar-related chromatin interactions.

Fig. R7 (A) Comparative analysis of interaction heatmaps between nHi-C and Hi-C. **(G)** Percentage of *cis/trans* A-A, A-B, and B-B compartment interactions captured by nHi-C and *in situ* Hi-C in HeLa cells. **(C-E)** Interaction heatmaps between chromosomes 16 and 17 in nHi-C **(C)**, *in situ* Hi-C **(D)** and *in situ* Hi-C after ActD treatment **(E)**. **(F-H)** hNADs interaction heatmaps between chromosomes 16 and 17 in nHi-C. **(F)** *in situ* Hi-C **(G)** and *in situ* Hi-C after ActD treatment **(H)**. **(I)** Scatter plot of hNADs trans interaction frequency before and after ActD treatment in Hi-C results. **(J)** Percentage of hNADs trans interaction detected before and after ActD treatment in Hi-C results.

Q2. The data obtained also confirmed the well-established from 3D-FISH results fact that nucleolus-associated interactions are enriched around centromeres. Please add the comparison of the obtained nHi-C data with previous 3D-FISH results to the discussion.

Response: Thanks for the reviewer’s suggestion, we added “Combined with the well-established 3D-FISH results that nucleolus-associated interactions are enriched around centromeres (Nemeth et al., 2010; van Koningsbruggen et al., 2010), this study depicts the genomic features and precise regions of hNADs involved.” in lines 347-350

page 17 in the discussion part.

Q3. It should be noted that rDNA-related interactions should be lower in NORs involved into nucleolus formation since inside the nucleolus rDNA is tightly surrounded by RNP aggregates. rDNA-related interactions could thus demonstrate different profiles in active and inactive NORs within the same cell line. Please add a discussion in this paragraph.

Response: Thanks for the reviewer's suggestion. We added "Since rDNA is tightly surrounded by RNP aggregates inside the nucleolus, rDNA-related interactions should be lower in NORs involved into nucleolus formation. We noticed that chromosomes 15, 21, 22 did not show much difference in interacting with rDNAs compared to chromosomes 13 and 14, while chromosome 14 showed the strongest interaction with rDNA." In line 328-332 page 16 in the discussion part.

Major comments:

C1. Lines 249-251 – Not only RNA polymerase I but also RNA polymerase II is sensitive to Actinomycin D treatment. Please indicate how these effects were separated and provide more details in the methods.

Response: Thanks for the reviewer's insightful suggestion. Actinomycin D (ActD) is a transcription inhibitor which intercalates into DNA and prevents the progression of RNA polymerases. According to the previous studies that ActD at low concentration (50 ng/ml) mainly affect RNAP I other than RNAP II (500 ng/ml) (Bensaude, 2011; Liu et al., 2016), we treated cells with low concentration ActD (10 ng /ml), which is proposed to specifically inhibit RNAPI. We also performed RNA-seq of HeLa cells treated with 10 ng/ml ActD treatment for 6 hours and 24 hours, to evaluate the effect of ActD at gene transcription level. 1799 differentially expressed genes (DEGs) were detected with 10 ng/ml ActD treatment for 6 hours and the number of DEGs increased to 3214 with 10

ng/ml ActD treatment for 24 hours (Fig. R8A-B). Combined with the immunostaining data of nucleolus with ActD treatment (Fig. R8C), it is reasonable to conclude that ActD treatment disassembles nucleolus at low concentration (10 ng/ml), which also have a side effect on RNAPII. We updated the experimental details in the method section in page 19, lines 377-380.

Fig. R8 (A-B) Volcano plot of differentially expressed genes in HeLa cells after 10 ng/ml ActD treatment for 6 hours (A) and 24 hours (B). **(C)** Immunostaining results of nucleolin in ActD non-treat (left) and ActD treated HeLa cells (right). (3D imaging results are provided as Supplementary Videos)

C2. Please provide additional evidence and references that centrifugation is enough to prepare nucleolus fraction.

Response: Thanks for the reviewer's valuable suggestion. The centrifugation strategy

used to prepare nucleolus fraction is a classical method inspired by many previous literatures (Amano, 1967; Busch et al., 1972; Yun Wah Lam, 2006). This strategy has also been used to extract nucleolus fraction to identify NADs in recent studies (Bizhanova et al., 2020; Dillinger et al., 2017; Pontvianne et al., 2016; Vertii et al., 2019). Besides the evidence of microscopic image of nucleoli after centrifugation and Western blotting results to validate the nucleolus extraction step (Figure S1 A-B, shown in Fig. R9A-B), seven hNAD regions identified from the nucleolus fraction and located in six chromosomes (Table R2) have been verified to closely adjacent to nucleolus by FISH experiments (Fig. R10), demonstrating the reliability of isolated nucleolus fraction.

Fig. R9 (A) Isolated nucleoli under microscopy. (B) Western blotting of nucleolar proteins from whole cells and isolated nucleoli.

Locus ID	Region (Mb)	Group
1	chr2:9.2-10.3	negative control
2	chr22:34-35.1	hNADs overlapped with previous study
3	chr15:53.5-55	hNADs outside of previous study
4	chr13:22.2-23.9	hNADs outside of previous study
5	chr15:56.6-56.9	HeLa-specific hNADs
6	chr12:22.8-24.2	
7	chr14:78.4-79.4	U2OS-specific hNADs
8	chr2:133.2-133.9	

Table R2 FISH target regions

C3. It would be beneficial to confirm hNAD identification and localization by 3D-FISH.

Response: Thanks for the reviewer's insightful suggestion. To validate the localization of hNADs, we selected seven representative hNADs (located in different

chromosomes, Fig. R10A-H) and one non-hNADs as negative control to perform DNA-FISH experiments. The results showed that all hNADs localized close to the nucleolus (Fig. R10I-K), which indicates the reliability of hNADs identified by nHi-C data. In addition, we compared hNADs in HeLa and U2OS cells. Two HeLa-specific hNADs and two U2OS-specific hNADs were selected and imaged by FISH experiments (Fig. R10E-H). It showed that HeLa-specific hNADs were tightly adjacent to the nucleolus in HeLa cells, but far away from the nucleolus in U2OS cells. On the contrary, U2OS-specific hNADs were tightly close to the nucleolus in U2OS cells, but far away from the nucleolus in HeLa cells (Fig. R10L-N). We added the figures in Fig. 2 and Fig. S4, and updated the manuscript in page 7 lines 146-164.

Fig. R10 (A-H) Track plot of genome regions chosen to perform DNA-FISH. **(I-K)** Representative images of hNADs' Oligopaint FISH probes in HeLa cells. **(L-N)** Representative images of HeLa/U2OS specific hNADs' Oligopaint FISH probes in HeLa and U2OS cells. (3D imaging results are provided as Supplementary Videos)

C4. Introduction lines 40-53 – By my opinion, this part of the text could be shortened because it describes the basic concepts of the nucleolus known from textbooks.

Response: Thanks for the reviewer's suggestion. We have shortened the descriptions of nucleoli in the introduction part (Page 3 lines 48-55 of manuscript).

Minor comments:

C1. Lines 155-162 – Please indicate which NORs are active and involved into nucleolus formation in the analyzed cell lines.

Response: Thanks for the reviewer's suggestion. We added "Our results indicated that NORs on chromosomes 15, 21, and 22 are more active and involved into

nucleolus formation in HeLa cells." in page 9 lines 189-190 of the manuscript.

Reference

- Amano, M. (1967). Metabolism of Rna in Liver Cells of Rat .I. Isolation and Chemical Composition of Nucleus Nucleolus Chromatin Nuclear Sap and Cytoplasm. *Exp Cell Res* 46, 169-&.
- Bensaude, O. (2011). Inhibiting eukaryotic transcription: Which compound to choose? How to evaluate its activity? *Transcription* 2, 103-108.
- Bizhanova, A., Yan, A., Yu, J., Zhu, L.J., and Kaufman, P.D. (2020). Distinct features of nucleolus-associated domains in mouse embryonic stem cells. *Chromosoma* 129, 121-139.
- Busch, H., Shibata, H., Yeoman, L.C., Rochoi, T.S., Reddy, R., Choi, Y.C., Daskal, I., Inagaki, A., and Olson, M.O.J. (1972). Isolation and Composition of Nuclei and Nucleoli. *Acta Endocrinol-Cop*, 35-+.
- Dillinger, S., Straub, T., and Nemeth, A. (2017). Nucleolus association of chromosomal domains is largely maintained in cellular senescence despite massive nuclear reorganisation. *PLoS One* 12, e0178821.
- Liu, X.F., Xiang, L.M., Zhou, Q., Carralot, J.P., Prunotto, M., Niederfellner, G., and Pastan, I. (2016). Actinomycin D enhances killing of cancer cells by immunotoxin RG7787 through activation of the extrinsic pathway of apoptosis. *P Natl Acad Sci USA* 113, 10666-10671.
- Nemeth, A., Conesa, A., Santoyo-Lopez, J., Medina, I., Montaner, D., Peterfia, B., Solovei, I., Cremer, T., Dopazo, J., and Langst, G. (2010). Initial genomics of the human nucleolus. *PLoS Genet* 6, e1000889.
- Pontvianne, F., Carpentier, M.C., Durut, N., Pavlistova, V., Jaske, K., Schorova, S., Parrinello, H., Rohmer, M., Pikaard, C.S., Fojtova, M., *et al.* (2016). Identification of Nucleolus-Associated Chromatin Domains Reveals a Role for the Nucleolus in 3D Organization of the *A. thaliana* Genome. *Cell Rep* 16, 1574-1587.
- van Koningsbruggen, S., Gierlinski, M., Schofield, P., Martin, D., Barton, G.J., Ariyurek, Y., den Dunnen, J.T., and Lamond, A.I. (2010). High-resolution whole-genome sequencing reveals that specific chromatin domains from most human chromosomes associate with nucleoli. *Mol Biol Cell* 21, 3735-3748.
- Vertii, A., Ou, J., Yu, J., Yan, A., Pages, H., Liu, H., Zhu, L.J., and Kaufman, P.D. (2019). Two contrasting classes of nucleolus-associated domains in mouse fibroblast heterochromatin. *Genome Res* 29, 1235-1249.
- Yun Wah Lam, A.I.L. (2006). Isolation of nucleoli. *Cell Biology (Third Edition) Chapter 15*, 103-107.

REVIEWERS' COMMENTS

Reviewer #1 (Remarks to the Author):

The authors provide a highly improved manuscript in response to the previous comments from the reviewers. The data clearly indicate that the "nHi-C" protocol described here is clearly superior to conventional Hi-C for the analysis of nucleolar-associated domains (NADs). This subject is of high interest to the rapidly expanding field of nuclear architecture and therefore I recommend publication.

Reviewer #2 (Remarks to the Author):

All my concerns have been addressed in the revised version of the manuscript entitled "Mapping Nucleolus-associated Chromatin Interactions Using Nucleolus Hi-C Reveals Pattern of Heterochromatin Interactions" and I can confidently recommend the manuscript for publication.